# Clinical Introduction of Stem Cell Sparing Radiotherapy to Reduce the Risk of Xerostomia in Patients with Head and Neck Cancer

**DOI:** 10.3390/cancers16244283

**Published:** 2024-12-23

**Authors:** Maria I. van Rijn-Dekker, Arjen van der Schaaf, Sanne W. Nienhuis, Antoinette S. Arents-Huls, Rachel B. Ger, Olga Hamming-Vrieze, Frank J. P. Hoebers, Mischa de Ridder, Sabrina Vigorito, Ellen M. Zwijnenburg, Johannes A. Langendijk, Peter van Luijk, Roel J. H. M. Steenbakkers

**Affiliations:** 1Department of Radiation Oncology, University Medical Centre Groningen, University of Groningen, 9700 RB Groningen, The Netherlands; m.i.van.rijn-dekker@umcg.nl (M.I.v.R.-D.); a.van.der.schaaf@umcg.nl (A.v.d.S.); s.w.nienhuis@umcg.nl (S.W.N.); j.a.langendijk@umcg.nl (J.A.L.); p.van.luijk@umcg.nl (P.v.L.); 2Department of Radiation Oncology, Radiotherapiegroep, 6815 AD Arnhem, The Netherlands; a.arents@radiotherapiegroep.nl; 3Radiation Oncology and Molecular Radiation Sciences, John Hopkins Medicine, Baltimore, MD 21287, USA; rger2@jhmi.edu; 4Department of Radiation Oncology, Netherlands Cancer Institute-Antoni van Leeuwenhoek, 1066 CX Amsterdam, The Netherlands; o.vrieze@nki.nl; 5Department of Radiation Oncology (MAASTRO), GROW School for Oncology and Reproduction, Maastricht University, 6229 ET Maastricht, The Netherlands; frank.hoebers@maastro.nl; 6Department of Radiation Oncology, University Medical Centre Utrecht, 3584 CX Utrecht, The Netherlands; m.deridder-5@umcutrecht.nl; 7Unit of Medical Physics, European Institute of Oncology IRCCS, 20141 Milan, Italy; sabrina.vigorito@ieo.it; 8Department of Radiation Oncology, Radboud University Medical Centre, 6525 GA Nijmegen, The Netherlands; e.zwijnenburg@radboudumc.nl

**Keywords:** head and neck cancer, xerostomia, parotid gland stem cells, stem cell sparing radiotherapy

## Abstract

Despite our efforts, head and neck cancer patients often experience xerostomia (i.e., the feeling of a dry mouth) after radiotherapy. Previous studies showed that dose to the parotid gland stem cells should be reduced to lower the risk of xerostomia. Our study aimed to bridge the gap between this research and clinical practice by assessing whether stem cell sparing radiotherapy can be used during treatment of head and neck cancer patients. Comparisons of radiation treatment plans demonstrated that radiation dose to the parotid gland stem cells can be reduced without jeopardizing the cancer treatment or the prevention of other side-effects. Moreover, a multicenter study showed that the clinical introduction of stem cell sparing radiotherapy only requires small adjustments of centers’ current optimization strategies. Taken together, stem cell sparing radiotherapy can now be used for head and neck cancer patients with the aim of reducing the risk of xerostomia after radiotherapy.

## 1. Introduction

Despite technological improvements [1,2], xerostomia remains frequently reported after radiotherapy (RT) for head and neck cancer (HNC) [3,4,5]. Recent studies indicated that a novel stem cell sparing (SCS)-RT technique may further reduce the risk of xerostomia [6,7,8,9,10,11,12]. Preclinical studies demonstrated that dose to a subvolume of the parotid gland, particularly rich in stem cells, was critical for preserving the gland’s functions after RT [6,7,8]. More specifically, dose to these stem cell rich (SCR) regions was associated with the deterioration of parotid gland saliva production and the development of xerostomia after RT [9,10]. Furthermore, mean dose (D_mean_) to the SCR regions (D_mean,SCR_) was a stronger risk factor for several xerostomia endpoints than D_mean_ to the parotid glands (D_mean,PG_) [10,11,12]. However, use of SCS-RT in clinical practice is still limited.

Gaps between research and change in clinical practice are well known [13,14,15]. Although various reasons exist for this (e.g., organizational, economic, and motivational), the outcome is the same: knowledge gained through scientific research is not effectively translated to clinical practice [13,14]. In addition, the clinical introduction of SCS-RT is probably hampered by the complexity of HNC treatment planning. Due to the anatomy in the head and neck region, dose reduction to a specified organ at risk (OAR) often results in increased dose to other OARs, especially when using photon-based technologies [16]. An optimal and generalizable SCS-RT strategy with currently used modern techniques (e.g., volumetric-modulated arc therapy (VMAT) and intensity-modulated proton therapy (IMPT)) is still lacking. Creating this type of strategy represents a first step to bridge the gap between research and change in practice [13]. More information regarding both the expected efficacy (i.e., achievable dose reduction in the SCR regions) and safety (i.e., target coverage and potential redistribution of dose to other OARs) is needed to provide insight into the application of SCS-RT. In addition, studies reported substantial variation in treatment plans between centers and improvement in OAR sparing after sharing best practice [17,18]. Therefore, reporting experiences in SCS-RT with routinely used RT technologies and from multiple centers can support effective and efficient clinical implementation by other institutes.

In conclusion, this study aims to assess whether SCS-RT can be adopted in routine clinical practice by the following methods:Providing recommendations;Quantifying the consequences with photon therapy (i.e., the expected benefit for the prevention of xerostomia and the potential effects for other side-effects);Testing the generalizability for other institutions;Exploring the added value for proton therapy.

## 2. Materials and Methods

### 2.1. Study Design

Multiple planning studies comparing SCS-RT with the current standard (ST-RT) were performed. In total, 30 HNC patients with squamous cell carcinoma tumors originating from the nasopharynx, hypopharynx, oropharynx, larynx, and oral cavity with varying TN-staging were selected to represent the general HNC population. All patients were scheduled for definitive radiotherapy with or without concurrent systemic treatment. Exclusion criteria were postoperative radiotherapy, previous head and neck irradiation, other tumor locations than those mentioned above, and no bilateral neck irradiation. The patients participated in the prospective data registration program (SFP, ClinicalTrials.gov number NCT02435576) of the department of Radiation Oncology at the University Medical Centre Groningen (UMCG) and gave informed consent for using their data in studies aimed at improving treatment.

#### 2.1.1. Step 1: Recommendations

To compare different SCS-RT strategies, 5 HNC patients were selected. For each patient, 4 SCS-RT plans (i.e., 2 with photons and 2 with protons) were made by 3 radiotherapy technologists (RTTs), in which SCR region dose was minimized while retaining target coverage and acceptable doses to other OARs. The strategies used and resulting SCS-RT plans were compared regarding SCR region dose reductions and unintentional dose shifts to other organs by an expert team consisting of RTTs, a radiation oncologist, a clinical physicist, and members of the research team. This resulted in planning recommendations for the application of SCS-RT, which were applied during the next steps.

#### 2.1.2. Step 2: Consequences with Photon Therapy

For 30 HNC patients with varying tumor locations and staging, SCS-RT plans were made using photon therapy. To determine the efficacy of SCS-RT, the differences in D_mean,SCR_ between the SCS-RT and ST-RT plans were calculated. To assess whether SCS-RT did not compromise target coverage or the overall sparing of normal tissues, the clinical goals (Appendix A) and the D_mean_ and V95 (i.e., volume receiving 95% of prescribed dose) for the whole body, as relative measures of integral dose and conformality, were inspected on a patient level. Additionally, changes in mean dose to other OARs (Appendix A) were calculated to detect unintentional dose shifts.

To estimate the clinical impact of the dose differences, the normal tissue complication probabilities (NTCPs) for several side-effects were compared between SCS-RT and ST-RT (Appendix A). To this end, previously published and validated NTCP models were used [3,10]. Based on the patient characteristics and the information from the RT plans (i.e., mean doses to several OARs), the NTCPs were calculated. The expected benefit of SCS-RT was estimated with NTCP models for several xerostomia endpoints, including patient-reported moderate-to-severe daytime and eating-related xerostomia and physician-reported grade ≥ 2 xerostomia [10]. The consequences of potential dose shifts to other OARs were estimated with NTCP models for a broad selection of side-effects, including physician-reported grade ≥ 2 dysphagia and patient-reported moderate-to-severe aspiration, xerostomia, sticky saliva, taste loss, speech problems, oral pain, nausea and vomiting, and fatigue [3]. The NTCP models from Van den Bosch et al. [3] were chosen since some of these are currently used in clinical practice in the Netherlands [19], while the NTCP models from van Rijn-Dekker et al. [10] are currently the only models incorporating the role of D_mean,SCR_. Moreover, those NTCP models were developed and validated following the same approach as the models from Van den Bosch et al. [3,10] As expected based on the multifactorial character of side-effects, these NTCP models did not only contain dosimetric predictors. For instance, the NTCP models for xerostomia did not only include D_mean,SCR_, but also other dose variables and pretreatment xerostomia complaints [10].

Considering the personalized character of RT, we explored explore which patients would benefit the most from SCS-RT. To this end, the relations between percentage reduction in SCR region dose and tumor location, minimum distance to target volume, and percentage part of SCR region outside target volume [20] were evaluated.

#### 2.1.3. Step 3: Generalizability for Other Institutions

A multicenter study was performed to test the applicability of SCS-RT in other institutes. In addition to the UMCG, 7 centers participated: John Hopkins Medicine, Netherlands Cancer Institute—Antoni van Leeuwenhoek Hospital, MAASTRO Clinic, Radiotherapiegroep Arnhem, University Medical Centre Utrecht, Istituto Europeo di Oncologia, and Radboud University Medical Centre. To test the hypothesis that each institute can achieve similar reductions in D_mean,SCR_ with photon therapy, 3 HNC patients were selected in whom substantial reductions of D_mean,SCR_ were observed during step 2. Each center, using their own treatment planning system (Appendix A), made ST-RT plans according to their own standard optimization strategy and SCS-RT plans by including objectives to reduce SCR region dose somewhere in their strategy. Recommendations regarding SCS-RT developed in step 1 were shared to support institutions.

#### 2.1.4. Step 4: Added Value for Proton Therapy

Proton therapy often reduces doses to the parotid glands and xerostomia [2,21]. Therefore, the potential additional benefit of SCS-RT for proton therapy was estimated. To this end, 15 oropharyngeal and nasopharyngeal cancer patients from step 2 were selected since these patients are known to generally benefit from proton therapy [22,23]. The reductions in D_mean,SCR_ and the consequences for target coverage and other OARs were evaluated in the same manner as during step 2.

### 2.2. Treatment Planning

VMAT was used for photon plans and IMPT for proton plans. All patients received the same RT fractionation schedule: 70 Gy in fractions of 2 Gy to the tumor volume and 54.25 Gy in fractions of 1.55 Gy to the elective volume using a simultaneous integrated boost technique. All patients received bilateral neck irradiation. The clinical goals for target coverage were based on Dutch guidelines (Appendix A). To reduce the risk of side-effects, OARs were contoured following published guidelines [24] and dose to OARs was specifically reduced during treatment planning (e.g., salivary glands, oral cavity, pharyngeal constrictor muscles [PCMs], brain structures). More details about the treatment planning systems used and optimization strategies can be found in Appendix A. All radiation plans were evaluated to be clinically acceptable and deliverable.

#### 2.2.1. Standard RT

In general, the clinical treatment plans were used as reference ST-RT plans. The exception is the multicenter study (step 3), in which new ST-RT plans were optimized according to centers’ own standard of practice.

#### 2.2.2. Stem Cell Sparing RT

The SCR regions were contoured according to published guidelines (see also Appendix A) [9,11]. Objectives to minimize SCR region dose were added in the optimization strategy (Appendix A). To minimize influence from the ST-RT plans, the SCS-RT plans were made from scratch (i.e., starting with an empty plan). In addition, RTTs responsible for the SCS-RT plans during steps 1, 2, and 4 were blinded for the ST-RT plans.

### 2.3. Statistical Analysis

To describe the differences in mean OARs dose and NTCPs between SCS-RT and ST-RT, descriptive statistics were calculated and presented as median (range). The Friedman test was used to test whether achieved dose differences varied between RTTs and centers (steps 1 and 3). Changes in mean dose to the SCR regions/other OARs and in NTCPs were evaluated using the Wilcoxon signed-rank test (steps 2 and 4). Spearman’s rho was used to correlate selected features with the percentage SCR region dose reduction (step 2). Differences were considered significant at *p* < 0.05. Analyses were performed using IBM Statistical Package for Social sciences version 28.

## 3. Results

In total, this study included 30 HNC patients (Table 1).

### 3.1. Step 1: Recommendations

By including objectives for the SCR regions, each RTT reduced D_mean,SCR_ with only limited dose shifts to other OARs (Appendix A). Despite using different optimization strategies, the reductions achieved in D_mean,SCR_ did not differ between RTTs (*p* > 0.36). However, RTT2 could not reduce D_mean,SCR_ in every patient. In addition, RTT1 also used separate objectives for the non-SCR regions, potentially explaining the decrease in D_mean,PG_ and D_mean_ to the non-SCR regions (D_mean,non-SCR_), especially with photon therapy. Finally, although dose shifts in other OARs were limited, differences between RTTs were noted (Appendix A), including an increase in D_mean_ to the inferior PCM by RTT2 (*p* = 0.03) and in D_mean_ to the brain by RTT1 (*p* = 0.04).

The comparison of the optimization strategies and resulting SCS-RT plans resulted in several observations and recommendations (Table 2). Most importantly, to add objectives for the SCR regions at the start of optimization and be aware of unintended dose increase (e.g., to the oral cavity or brain).

### 3.2. Step 2: Consequences with Photon Therapy

Overall, median D_mean,SCR_ reductions of 4.1 and 3.5 Gy were achieved for ipsilateral and contralateral, respectively (*p* < 0.001, Figure 1, Appendix A). Mean doses to the parotid glands and non-SCR regions were also reduced (*p* < 0.001), since objectives for all parotid gland structures were included during optimization based on the recommendations (Table 2). At a population level, the estimated clinical impact on several xerostomia outcomes was significant (*p* < 0.001, Appendix A). More specifically, if ≥2% NTCP reduction is considered clinically relevant, SCS-RT was predicted to decrease daytime xerostomia, eating-related xerostomia, and physician-rated grade ≥ 2 xerostomia in 6 (20%), 4 (13%), and 14 (47%) of the patients, respectively (Figure 2).

Dose reductions in the SCR regions were accomplished with similar target coverage, also near the parotid glands, and similar overall sparing of normal tissues (Appendix A). The increases in D_mean_ to the brain and brainstem were considered clinically irrelevant (median 0.0 and 0.3 Gy, respectively, Appendix A). At a population level, no significant increase in dose to other OARs was observed (Figure 1B and Appendix A). However, at a patient level, D_mean_ oral cavity (D_mean,oral_) increased with ≥2 Gy in six patients, emphasizing the relevance of our fourth recommendation (Table 2). On the other hand, doses to the submandibular glands and supraglottic area were significantly lower with SCS-RT (*p* < 0.02). At a population level, the observed dose shifts did not significantly increase the NTCPs for other side-effects (Appendix A). Moreover, the NTCPs for xerostomia, sticky saliva, taste loss, and speech problems were significantly lower (*p* < 0.004) compared to ST-RT. For example, at a patient level, 17 (57%) patients were predicted to experience a ≥ 2% reduction in general xerostomia (Appendix A). This was explained by dose reductions in the salivary glands and supraglottic area (i.e., the model predictors). On the other hand, the risk of dysphagia was increased by ≥2% in three (10%) patients (Appendix A), which was due to an increased D_mean,oral_.

More D_mean,SCR_ reduction was possible in nasopharyngeal cancer patients compared to HNC patients with different tumor locations (rho = −375, Appendix A). In addition, the largest dose reductions were observed if the minimum distance between the SCR regions and elective target volume was larger (rho = −0.351, Appendix A) or if the minimum distance between the SCR regions and tumor target volumes was intermediate (rho = −0.448, Appendix A). In line with this, less overlap between the SCR regions and the target volumes correlated with more D_mean,SCR_ reduction (Appendix A).

### 3.3. Step 3: Generalizability for Other Institutions

By making only small adjustments to their own optimization strategy, all institutes, each with their own treatment planning system, could reduce D_mean,SCR_ without compromising target coverage or unacceptable dose shifts to other OARs (Figure 3 and Figure 4). However, the achieved dose reduction in the contralateral SCR region differed between the centers (*p* = 0.01, Appendix A). This was likely due to substantial differences in the ST-RT plans (Figure 3 and Appendix A). The application of SCS-RT planning reduced the differences in SCR region doses between institutes.

In general, dose shifts to other OARs due to SCS-RT were limited (Appendix A). Moreover, no significant differences were observed between the centers, except for D_mean_ to the brainstem (*p* = 0.05). However, these were considered not clinically relevant (ranging from −0.5 to 0.9 Gy, Appendix A).

### 3.4. Step 4: Added Value for Proton Therapy

Although proton therapy already results in lower D_mean,PG_ (Appendix A) [2,21,25], SCS-RT further reduced D_mean,SCR_ (*p* < 0.002, Figure 1A and Appendix A). Median D_mean,SCR_ reductions of 2.2 and 1.9 Gy for ipsilateral and contralateral, respectively, were achieved without compromising target coverage and overall sparing of normal tissues (Appendix A). However, SCS-RT did result in a small but significant increase in D_mean,oral_ (median increase = 0.2 Gy, *p* = 0.01, Appendix A). At a population level, SCS-RT did not reduce NTCPs for xerostomia outcomes (*p* > 0.17, Appendix A). Overall, a ≥2% NTCP benefit for xerostomia was seen in very few patients, only for general xerostomia in two (13%) patients. A potential reason for this observation was the already low D_mean,SCR_ in the ST-RT proton plans, which was already quite comparable with the D_mean,SCR_ in the SCS-RT photon plans (Appendix A). In addition, the dose shifts to other OARs did not result in significant changes of NTCPs for other side-effects (Appendix A).

Taken together, this study demonstrated that SCS-RT can be implemented in daily practice by slightly adjusting a center’s current clinical standard. Using the aforementioned recommendations, D_mean,SCR_ can be reduced with both photon and proton therapy.

## 4. Discussion

The current study bridged the gap between studies showing the expected benefit of SCS-RT [6,7,8,9,10,11,12], and the clinical introduction of SCS-RT. We provided recommendations to apply SCS-RT and showed that SCR region dose can be reduced without compromising target coverage or increasing the risk of other side-effects. Nevertheless, awareness of unintended increase of dose to other OARs during treatment optimization, especially to the oral cavity, remains pivotal. This novel technique can contribute to the prevention of xerostomia in HNC patients treated with photon therapy. Lastly, the multicenter study demonstrated the generalizability of SCS-RT.

### 4.1. Towards Personalized Radiotherapy

Using photon therapy, D_mean,SCR_ was decreased. The achieved dose reductions were comparable with previous studies testing SCS-RT, despite the differences in delineation strategy of the parotid gland stem cells, fractionation schedules, and optimization strategies [11,12]. Moreover, the lower NTCPs for several xerostomia outcomes in the SCS-RT plans were in line with the significantly lower xerostomia incidence in patients treated with SCS-RT as reported by Fried et al. and Huang et al. [12,26] Taken together, SCS-RT is expected to have a clinically relevant impact in the prevention of radiation-induced xerostomia.

Although D_mean,SCR_ was already reduced by proton therapy in the ST-RT plans (Appendix A), SCS-RT further decreased this. Unfortunately, this did not translate into a clinically relevant reduction in the NTCPs for several xerostomia outcomes. The limited reductions in D_mean,SCR_ and the small increases in D_mean,oral_ indicated a dose redistribution between OARs in the SCS-RT plans. Since both D_mean,SCR_ and D_mean,oral_ are predictors in the NTCP models used for xerostomia outcomes [10], this probably explained the similar NTCPs with ST-RT and SCS-RT. However, this might change with upcoming new technologies (e.g., proton arc therapy [27] or spot size reduction for intensity-modulated proton therapy [28]).

Since SCS-RT can easily be implemented, this technique should be recognized as low-hanging fruit to reduce the risk of xerostomia in HNC patients treated with photon therapy. This is especially true since not all HNC patients will be able to benefit from the often more favorable dose distributions with proton therapy. Although the reduction in NTCP for xerostomia varies, substantial NTCP reductions were achieved in several patients (Figure 2 and Appendix A). To support personalized RT treatment, the current study also characterized patients who will benefit the most from SCS-RT. Larger dose reductions in the SCR regions were observed in nasopharyngeal cancer patients, which is a subgroup of patients in whom sparing the whole parotid glands was difficult [29].

### 4.2. Safety: Oncological and Toxicity Outcomes

Although SCS-RT did not affect target coverage, the current study was not designed to assess the impact on oncological outcomes. However, no difference in oncological outcomes is expected since target coverage according to the Dutch guidelines was met in all SCS-RT plans. Moreover, an analysis on the survival data from our previous randomized controlled trial indicated that SCS-RT did not influence patterns of failure. More specifically, using this cohort of 102 HNC patients, we observed no difference in the overall survival after almost 5 years [30]. In addition, locoregional failure near the parotid glands was proven in only seven patients (7%) and always occurred in the high-dose area, which was never compromised during SCS-RT [30].

In line with previous studies, reduction in D_mean,SCR_ was mostly possible without jeopardizing sparing of other OARs [11,12,26]. Overall, on a population level, no clinically relevant increase in NTCPs for other toxicities were observed. Moreover, NTCPs for other toxicities in the salivary domain (i.e., general xerostomia, sticky saliva, and taste loss) were lowered. This was primarily explained by the reduction in D_mean, PG_ due to the inclusion of objectives for all parotid gland structures during SCS-RT (Table 2). However, awareness of unintended dose increase, especially to the oral cavity, is warranted during treatment optimization since this can result in higher NTCPs for dysphagia. Since xerostomia is not the only side-effect to prevent, this emphasizes the need to search for an acceptable balance when sparing OARs during treatment optimization, for example by using NTCP models during treatment optimization [19,31,32,33].

### 4.3. Implementation in Your Own Center

This multicenter study showed that other institutes could apply SCS-RT by making small adjustments to their current planning strategy. In line with the literature [17,18], many differences were noted between the centers, amongst others, different treatment planning systems (i.e., RayStation, Eclipse, Pinnacle, and Monaco), different optimization strategies (e.g., manual vs. semi-automated and optimization based on dose volume histogram points, mean dose, and equivalent uniform dose), and different priorities regarding sparing OARs (e.g., salivary OARs or swallowing OARs) (Appendix A). This resulted in different mean OARs’ doses (Appendix A). However, each center could reduce D_mean,SCR_ with their own treatment planning system and optimization strategy after receiving the recommendations developed in step 1. An additional effect of the planning assessment from the multicenter study was more similar D_mean,SCR_ in the SCS-RT plans (Appendix A). This was in accordance with Verbakel et al., showing the improvement of OAR sparing after targeted intervention [18]. In addition, despite the use of various treatment planning systems (Appendix A), each institution was able to achieve lower D_mean,SCR_. This was in line with others demonstrating that different treatment planning systems can achieve similar treatment plan quality [34,35]. Moreover, to further facilitate the implementation of SCS-RT, a protocol for the delineation of the SCR regions is provided in Appendix A Lastly, the implementation of a new planning strategy requires time (e.g., adjusting protocols, training staff, and evaluating the changes made). From our own experience and the multicenter study, the time investment to adopt SCS-RT in your own standard of practice is limited since it is only the addition of one OAR (i.e., the SCR regions).

### 4.4. Limitations

The use of manually made treatment plans probably induced additional variation in the obtained results. For instance, observed dose differences could have been affected by inter-RTT variation. This issue might be solved by using automated treatment planning [35,36,37]. However, to properly integrate SCS-RT in automated treatment planning, more information about SCS-RT is needed since automated treatment planning is based on a library of clinically acceptable treatment plans or a priori defined clinical goals [37,38,39,40]. In the future, multiple SCS-RT plans can be added to the library of treatment plans. Subsequently, the models behind the automated treatment planning can be trained again to eventually allow automated SCS-RT planning. Furthermore, plan comparison studies can be at risk of being an unfair representation of reality. Although the number of patients used in the study is low, we have tried to minimize this risk by mimicking clinical practice: (1) patients were consecutively included per tumor location to avoid biased patient selection, (2) RTTs were instructed to aim for a clinically acceptable and deliverable plan within similar time constraints as in clinical practice, and (3) RTTs responsible for the SCS-RT plans used for providing recommendations (step 1) and evaluating the consequences (step 2 and 4) were blinded for the ST-RT plans since such plans are also not available during clinical practice.

For quality of life, many side-effects play a role, not only xerostomia. Xerostomia is still one of the most reported side-effects after radiotherapy [3]. So, we believe that reducing the chance of developing xerostomia will eventually have a positive impact on quality of life.

## 5. Conclusions

By making only small adjustments to the local optimization strategies, stem cell sparing radiotherapy can easily be implemented. This novel technique can reduce the risk of xerostomia in HNC patients treated with photon therapy without jeopardizing target coverage and the prevention of other side-effects. However, the clinical relevance for patients treated with proton therapy is currently limited.

## Figures and Tables

**Figure 1 cancers-16-04283-f001:**
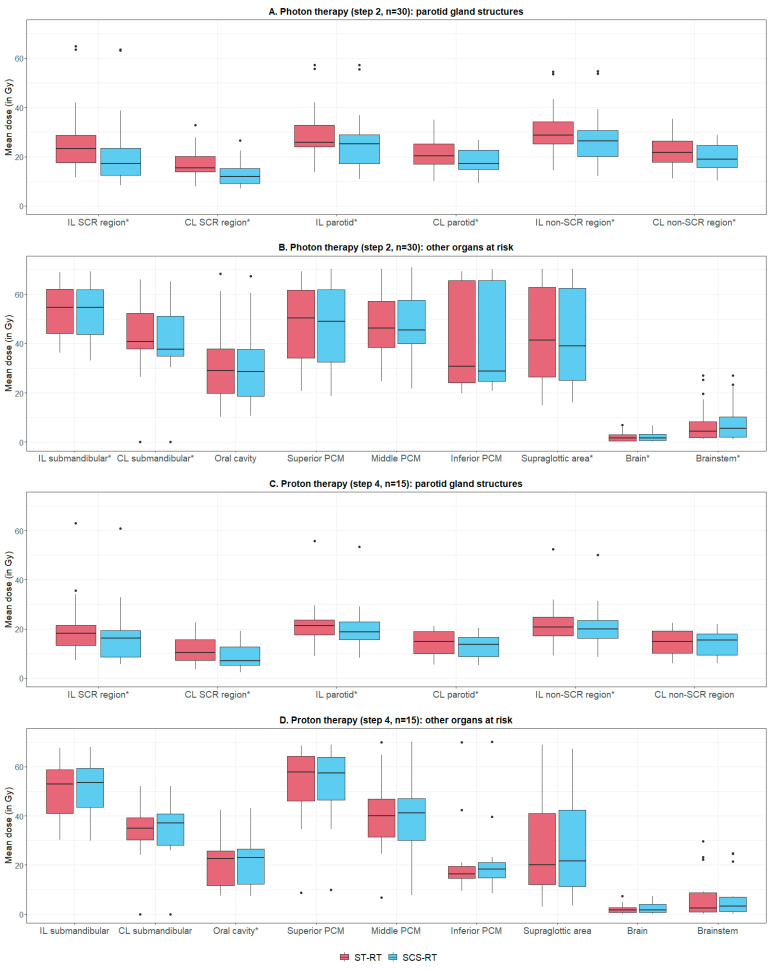
Dose shifts with photon (step 2) and proton therapy (step 4). Abbreviations: CL = contralateral (i.e., receiving lowest dose); IL = ipsilateral (i.e., receiving highest dose); non-SCR = remaining parotid gland tissue (i.e., parotid gland minus SCR region); PCM = pharyngeal constrictor muscle; RT = radiotherapy; SCR = stem cell rich; SCS = stem cell sparing; ST = standard. * Significant according to Wilcoxon signed-rank test (significance level of *p* < 0.05).

**Figure 2 cancers-16-04283-f002:**
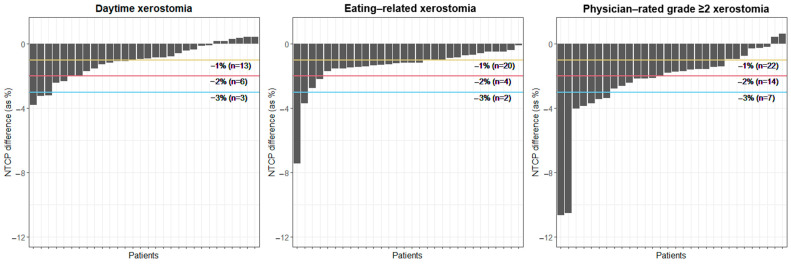
Estimated clinical impact of SCSRT with photon therapy on several xerostomia outcomes. This figure depicts the NTCP differences (i.e., NTCP in ST-RT plan minus NTCP in SCS-RT plan) for several xerostomia outcomes. The NTCPs were calculated using the models developed by van Rijn-Dekker et al. [10]. The lines depict the number of patients in which the NTCP was decreased by at least 1% (yellow), 2% (red), and 3% (blue). Abbreviations: NTCP = normal tissue complication probability; SCS = stem cell sparing; ST = standard; RT = radiotherapy.

**Figure 3 cancers-16-04283-f003:**
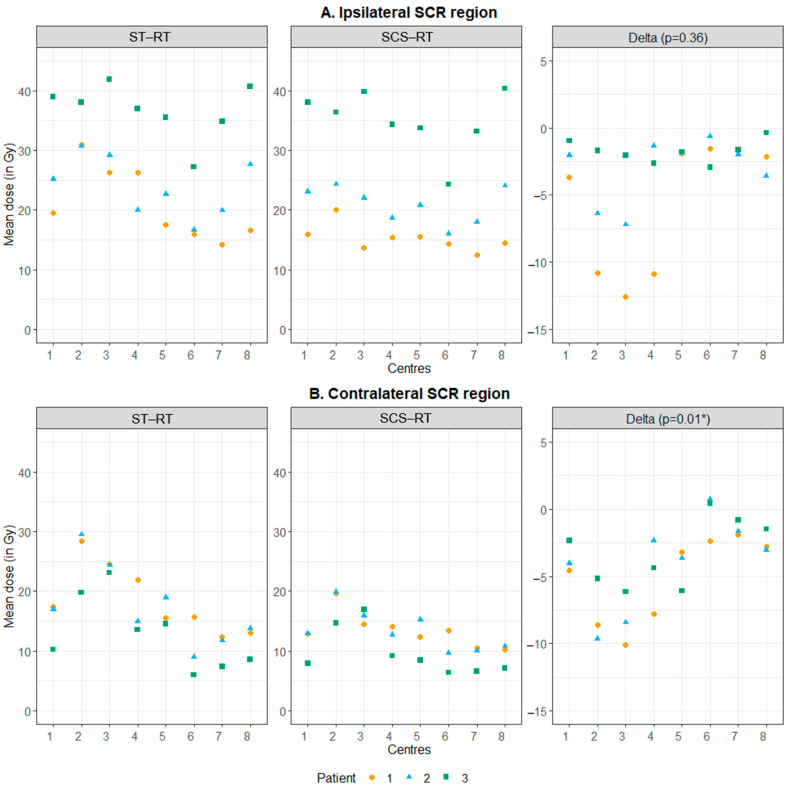
Multicenter study (step 3): achieved dose reductions in the SCR regions. Abbreviations: RT = radiotherapy; SCR = stem cell rich; SCS = stem cell sparing; ST = standard. * Significant according to Friedman test (significance level of p < 0.05)

**Figure 4 cancers-16-04283-f004:**
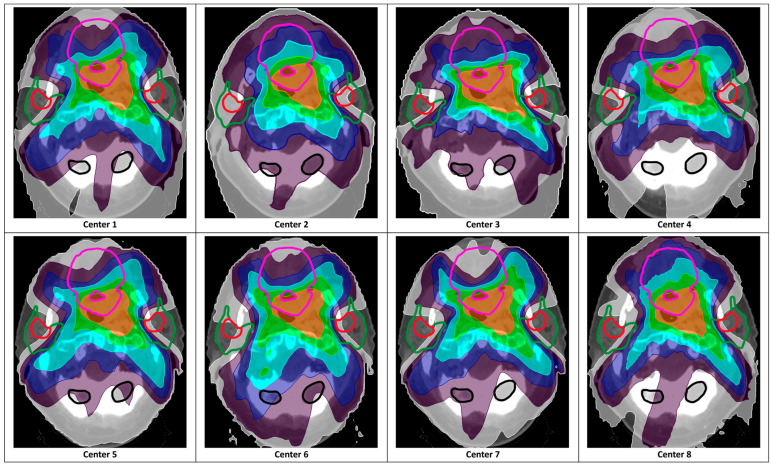
Application of SCS-RT by slightly adjusting the centers’ own standard treatment optimizations. Examples from the different dose distributions of the SCS-RT plan from the different centers (orange = 66.5 Gy; green = 51.54 Gy; light blue = 40 Gy; dark blue = 30 Gy; purple = 20 Gy; white = 10 Gy) demonstrating that each center was able to reduce mean SCR region dose. The following OARs were delineated: SCR regions (red), parotid glands (green), oral cavity (pink), and brain (black). Abbreviations: Gy = Gray; RT = radiotherapy; SCR = stem cell rich; SCS = stem cell sparing.

**Table 1 cancers-16-04283-t001:** Patient characteristics per cohort used during the different steps.

	Recommendations for SCS-RT (Step 1, n = 5)	Consequences with Photon Therapy (Step 2, n = 30)	Generalizability for Other Institutes (Step 3, n = 3)	Added Value for Proton Therapy (Step 4, n = 15)
Sex, No. (%)				
Female	2 (40)	11 (37)	1 (33)	5 (33)
Male	3 (60)	19 (63)	2 (67)	10 (67)
Age, median (range), y	68 (42–76)	69 (36–85)	68 (66–73)	65 (42–77)
Tumor location, No. (%)				
Larynx	0 (0)	5 (17)	0 (0)	0 (0)
Hypopharynx	0 (0)	5 (17)	0 (0)	0 (0)
Oropharynx	3 (60)	9 (30)	3 (100)	9 (30)
Nasopharynx	2 (40)	6 (20)	0 (0)	6 (20)
Oral cavity	0 (0)	5 (16)	0 (0)	0 (0)
Tumor stage, No. (%)				
T1–2	2 (40)	11 (37)	0 (0)	6 (40)
T3–4	3 (60)	19 (63)	3 (100)	9 (60)
Nodal stage, No. (%)				
N0	1 (20)	4 (13)	0 (0)	2 (13)
N+	4 (80)	26 (87)	3 (100)	13 (87)
CTV volume, median (range), cm^3^				
High dose (70 Gy)	66 (36–123)	69 (9–343)	66 (44–77)	66 (21–263)
Low dose (54.25 Gy)	321 (199–404)	338 (137–601)	295 (290–320)	327 (196–599)
PTV volume, median (range), cm^3^				
High dose (70 Gy)	103 (64–199)	117 (20–517)	103 (85–124)	103 (37–411)
Low dose (54.25 Gy)	534 (350–632)	551 (259–882)	467 (463–534)	537 (350–882)

Abbreviations: CTV = clinical target volume; PTV = planning target volume; RT = radiotherapy; SCS = stem cell sparing.

**Table 2 cancers-16-04283-t002:** Observations and recommendations of the optimization strategies.

**Observations**
When objectives for the SCR regions were added in a fully optimized plan, dose reductions were limited. The inclusion of objectives for the non-SCR regions reduced mean dose to the non-SCR regions and the parotid glands as well, especially with photon therapy. The removal of objectives for the whole parotid glands did not further reduce mean SCR region dose but increased mean parotid gland dose. The use of out-structures as proposed by Tambas et al. [20] resulted in larger dose reductions in the SCR regions. When the SCR regions were prioritized, dose redistribution to other OARs (especially the oral cavity and the brain) and to the PTV_54.25_ (resulting in hotspots) was observed.
**Recommendations**
Add objectives for the SCR regions at the start of optimization. Use objectives for all parotid gland structures (i.e., whole parotid gland, SCR regions, and non-SCR regions) to optimally optimize dose and have more control. Use out-structures of the SCR regions, in which dose can be reduced in the OAR part outside the PTV54.25 expanded by a 5 mm margin [20]. Be aware of unintended dose increase, especially to the oral cavity, brain, and PTV54.25.

Abbreviations: non-SCR = remaining parotid gland tissue (i.e., parotid gland minus SCR region); PTV_54.25_ = elective dose (54.25 Gy) planning target volume; RT = radiotherapy; SCR = stem cell rich; SCS = stem cell sparing.

## Data Availability

The raw data supporting the conclusions of this article will be made available by the authors on request.

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
