# Peer review of "Clinical Introduction of Stem Cell Sparing Radiotherapy to Reduce the Risk of Xerostomia in Patients with Head and Neck Cancer"

_cancers, 2024, doi:10.3390/cancers16244283_

Round 1

Reviewer 1 Report

Comments and Suggestions for Authors

1. Insufficient Impact on Clinical Outcomes

  • While the study demonstrates reductions in mean dose to stem cell-rich (SCR) regions (Dmean,SCR), the clinical impact, particularly for proton therapy, is limited. For instance, the reduction in normal tissue complication probabilities (NTCPs) for xerostomia in patients treated with proton therapy is marginal and statistically insignificant (p > 0.17). This weakens the case for adopting stem cell sparing (SCS) radiotherapy in proton therapy settings.

2. Unclear Applicability in Real-World Settings

  • Despite showcasing the feasibility of SCS-RT in a multicenter setup, the article lacks details on the challenges encountered during implementation across centers. Variability in treatment planning systems and optimization protocols introduces uncertainties that are not thoroughly explored.

3. Limited Focus on Long-Term Outcomes

  • The study emphasizes NTCP reductions for xerostomia without addressing other significant long-term outcomes, such as overall survival, quality of life, or comprehensive oncological effectiveness.

4. Potential Dose Redistribution Issues

  • The study acknowledges unintended dose increases to other organs at risk (OARs), such as the oral cavity and pharyngeal constrictor muscles, which might lead to increased risks of dysphagia and other toxicities. However, these implications are not fully quantified or discussed in depth.

5. Inconsistent Benefit Across Patient Subgroups

  • The benefits of SCS-RT are not consistent across all head and neck cancer subgroups. For instance, nasopharyngeal cancer patients showed greater dose reductions than others, raising questions about the universal applicability of the proposed approach.

6. Reliance on Manual Treatment Planning

  • Manual treatment planning introduces variability in results due to inter-operator differences. Although this issue is acknowledged, no robust solution, such as automated treatment planning systems, is discussed for standardizing results.

7. Lack of Transparency in Methodological Details

  • Key methodological details, such as how NTCP models were adapted for different centers and tumor types, are not described in sufficient detail. This reduces the reproducibility of the study.

8. Limited Sample Size

  • The study includes only 30 patients for photon therapy and 15 for proton therapy in stepwise evaluations. Such a small sample size may not adequately represent the diversity of clinical scenarios, limiting the generalizability of the findings.

9. No Validation Against Deep Learning Approaches

  • The study relies on traditional NTCP modeling and treatment planning without exploring advanced machine learning or deep learning methods, which might provide better predictive accuracy and treatment optimization.

10. Bias from Center-Specific Practices

  • Variability in baseline standard radiotherapy (ST-RT) plans across centers introduces bias, as some centers may already employ optimization strategies that partially overlap with SCS-RT, artificially inflating observed improvements.

11. Limited Assessment of Patient-Reported Outcomes

  • The study heavily relies on NTCP models to estimate xerostomia risk without incorporating robust patient-reported outcomes or real-world data on quality of life, which could provide a more holistic view of the benefits.

12. Lack of Emphasis on Economic and Operational Feasibility

  • While the article claims that SCS-RT requires only small adjustments, it does not analyze the economic or operational feasibility of implementation, such as training requirements for staff or costs associated with updated protocols.
Comments on the Quality of English Language

The English could be improved to more clearly express the research.

Author Response

  1. Insufficient Impact on Clinical Outcomes

While the study demonstrates reductions in mean dose to stem cell-rich (SCR) regions (Dmean,SCR), the clinical impact, particularly for proton therapy, is limited. For instance, the reduction in normal tissue complication probabilities (NTCPs) for xerostomia in patients treated with proton therapy is marginal and statistically insignificant (p > 0.17). This weakens the case for adopting stem cell sparing (SCS) radiotherapy in proton therapy settings.

Response: Indeed, we agree with the Reviewer that the current study did not show an additional benefit for stem cell sparing (SCS) radiotherapy in head and neck (HNC) patients treated with proton therapy (lines 331-332). However, this might change in the future with newer techniques, like proton arc (lines 336-338). Nevertheless, proton therapy is not available for all HNC patients. Consequently, the statistically significant and clinically relevant normal tissue complication probability (NTCP) reduction for xerostomia with photon therapy is very valuable.

We have emphasized this in the Discussion:

Line 341-342: “Especially since not all HNC patients will be able to benefit from the often more favorable dose distributions with proton therapy.”

  1. Unclear Applicability in Real-World Settings

Despite showcasing the feasibility of SCS-RT in a multicenter setup, the article lacks details on the challenges encountered during implementation across centers. Variability in treatment planning systems and optimization protocols introduces uncertainties that are not thoroughly explored.

Response: Indeed, in line with literature (Verbakel et al. 2019), we found many differences between centers regarding the treatment planning systems, optimization protocols, organs at risk of interest, and so on. However, despite all these differences, each center was able to adopt SCS radiotherapy.

We have elaborated a bit more about this in the Discussion:

Lines 371-379: In line with literature[17,18], many differences were noted between the centers, amongst others, different treatment planning systems (i.e., RayStation, Eclipse, Pinnacle, and Monaco), different optimization strategies (e.g., manual vs. semi-automated and optimization based on dose volume histogram points, mean dose, and equivalent uniform dose), and different priorities regarding sparing OARs (e.g., salivary OARs or swallowing OARs) (Table S4). This resulted in different mean OARs doses (Table S10). However, each center could reduce Dmean,SCR with their own treatment planning system and optimization strategy after receiving the recommendations developed in step 1.

  1. Limited Focus on Long-Term Outcomes

The study emphasizes NTCP reductions for xerostomia without addressing other significant long-term outcomes, such as overall survival, quality of life, or comprehensive oncological effectiveness.

Response: The main objective of a radiotherapy treatment is to treat the tumor. Therefore, no concessions were made regarding the tumor coverage in the SCS radiotherapy plans. The target coverage according to current Dutch guidelines was met in all treatment plans (lines 251-252). Consequently, no differences in oncological outcomes were expected. We have added this to the Discussion:

Lines 350-353: However, no difference in oncological outcomes is expected since target coverage according to the Dutch guidelines was met in all SCS-RT plans. Moreover, an analysis on the survival data from our previous randomized controlled trial indicated that SCS-RT did not influence patterns of failure.

In addition, we agree with the Reviewer that xerostomia is only a part of quality of life. Nevertheless, if it is possible to reduce the risk of such a relevant and prevalent side-effect as xerostomia, we should optimize our plans accordingly.

  1. Potential Dose Redistribution Issues

The study acknowledges unintended dose increases to other organs at risk (OARs), such as the oral cavity and pharyngeal constrictor muscles, which might lead to increased risks of dysphagia and other toxicities. However, these implications are not fully quantified or discussed in depth.

Response: The Reviewer is right, due to the anatomy in the head and neck region, reduction to a specified organ at risk (OAR) often results in increased dose to other OARs, especially when using photon-based technologies. Since this is often inevitable, this study aimed to gain more information about the consequences of redistribution of the dose. Therefore, we investigated the effects of SCS radiotherapy on other side-effects than xerostomia, like dysphagia and fatigue.

We emphasized this in the study aim in the Introduction:

Lines 97-98: 2. quantifying the consequences with photon therapy (i.e., the expected benefit for the prevention of xerostomia and the potential effects for other side-effects);

In addition, we added this to the Discussion:

Lines 365-368: Since xerostomia is not the only side-effect to prevent, this emphasizes the need to search for an acceptable balance when sparing OARs during treatment optimization, for example by using NTCP models during treatment optimization.[30–33]

  1. Inconsistent Benefit Across Patient Subgroups

The benefits of SCS-RT are not consistent across all head and neck cancer subgroups. For instance, nasopharyngeal cancer patients showed greater dose reductions than others, raising questions about the universal applicability of the proposed approach.

Response: Our study aimed to assess the added value and consequences of SCS radiotherapy in the general HNC population. Therefore, 30 HNC patients were selected as representatives of the general HNC population. However, radiotherapy is always an personalized treatment. Consequently, we aimed to gain more insight into the benefits and consequences of SCS radiotherapy for individual patients (i.e., the subgroup analysis as described in lines 151-154). We believe this information is valuable since it shows which patients will specifically benefit from this new technique.

We have elaborated on the patient selection in the Materials and Methods to show that the study population aimed to be a representation of the general HNC population:

Lines 104-109: In total, 30 HNC patients with squamous cell carcinoma tumors originating from the nasopharynx, hypopharynx, oropharynx, larynx and oral cavity with varying TN-staging were selected to represent the general HNC population. All patients were scheduled for definitive radiotherapy with or without concurrent systemic treatment. Exclusion criteria were postoperative radiotherapy, previous head and neck irradiation, other tumor locations than those mentioned above, and no bilateral neck irradiation.

Moreover, we have further emphasized the personalization of radiotherapy treatment:

Lines 151-154: Considering the personalized character of RT, we explored which patients would benefit the most from SCS-RT. To this end, the relations between percentage reduction in SCR region dose and tumor location, minimum distance to target volume, and percentage part of SCR region outside target volume[19] were evaluated.

Lines 344-345: To support personalized RT treatment, the current study also characterized patients who will benefit the most from SCS-RT.

  1. Reliance on Manual Treatment Planning

Manual treatment planning introduces variability in results due to inter-operator differences. Although this issue is acknowledged, no robust solution, such as automated treatment planning systems, is discussed for standardizing results.

Response: We have added our thoughts towards a robust solution to the Discussion:

Lines 399-401: In the future, multiple SCS-RT plans can be added to the library of treatment plans. Subsequently, the models behind the automated treatment planning can be trained again to eventually allow automated SCS-RT planning.

  1. Lack of Transparency in Methodological Details

Key methodological details, such as how NTCP models were adapted for different centers and tumor types, are not described in sufficient detail. This reduces the reproducibility of the study.

Response: We agree with the Reviewer that more details about the application of the NTCP models can be given. We have used previously published and validated NTCP models. These models were not adjusted for the current study. To predict the occurrence of xerostomia and other side-effects, patient characteristics and information from the RT plans (i.e., mean doses to several OARs) were derived from the patient files and treatment plans and used to calculate the different NTCPs.

The NTCP models from van Rijn-Dekker et al. (2023) were used to estimate the benefit of SCS radiotherapy, since these are the only available models considering the role of the stem cell rich regions. Thet NTCP models from Van den Bosch et al. (2021) were used to investigate the consequences of SCS radiotherapy for other side-effects. These models were chosen because they are also currently used in clinical practice in the Netherlands (i.e., for selection of proton therapy (Langendijk et al. 2021) and treatment optimization). Moreover, the NTCP models from van Rijn-Dekker et al. (2023) were developed and validated following the same strategy as the NTCP models from Van den Bosch et al. (2021).

We have added this to the Materials and Methods:

Lines 143-147: The NTCP models from Van den Bosch et al. [3] were chosen since some of these are currently used in clinical practice in the Netherlands [33], while the NTCP models from van Rijn-Dekker et al. [10] are currently the only models incorporating the role of Dmean,SCR. Moreover, those NTCP models were developed and validated following the same approach as the models from Van den Bosch et al. [3,10]

  1. Limited Sample Size

The study includes only 30 patients for photon therapy and 15 for proton therapy in stepwise evaluations. Such a small sample size may not adequately represent the diversity of clinical scenarios, limiting the generalizability of the findings.

Response: We agree with reviewer that the number of included patients is quite low. Still, the study cohort was aimed to represent the general HNC population (please review the answer to question 5 for more details). The amount of dose reduction in the SCR region is quite different among the included 30 patients and also different among the patients in the different subgroups (nasopharynx, oropharynx, hypopharynx, larynx and oral cavity) as can be depicted in figures S3 and S4 in the supplementary materials. Therefore, we believe that including more patients to the study would not have likely changed these findings and subsequently alter the generalizability of the study.

We added the low number of patients to the limitations in the Discussion:

                   Lines 402-403: Although the number of patients used in the study is low, we have tried to minimize this risk by mimicking clinical practice

  1. No Validation Against Deep Learning Approaches

The study relies on traditional NTCP modeling and treatment planning without exploring advanced machine learning or deep learning methods, which might provide better predictive accuracy and treatment optimization.

Response: Indeed, we used traditional NTCP models and manual treatment plans. First, we have chosen to use the NTCP models that are currently used in clinical practice in the Netherlands (Langendijk et al., 2021) and the only NTCP models currently available that address the role of the SCR regions. All these models were validated, which is not yet the case for all deep learning prediction models. Moreover, we aimed to assess the impact on several side-effects. Currently, no deep learning models are available for all those side-effects. Please review the answer to question 7 for more details about the NTCP models used and the changes made to the Manuscript to clarify this.

Second, the radiotherapy treatment plans were manually made since SCS radiotherapy is not yet integrated in the machine learning models used for automated treatment planning. Please review the answer to question 6 for our suggested approach to adopt SCS radiotherapy in automated treatment planning and the changes made to the Manuscript accordingly.

  1. Bias from Center-Specific Practices

Variability in baseline standard radiotherapy (ST-RT) plans across centers introduces bias, as some centers may already employ optimization strategies that partially overlap with SCS-RT, artificially inflating observed improvements.

Response: We agree with the Reviewer that each center uses its own treatment optimization process and often different treatment planning systems. We have made changes in the Manuscript to further emphasize this (please review the answer to question 2 for the changes made). This was also one of the reasons to analyze the differences in mean doses to OARs, instead of only the mean doses in the SCS radiotherapy plans. Lastly, in line with Verbakel et al. (2019), we showed that the variation in OAR sparing became smaller due to a planning assessment, like our multicenter study. We have emphasized this more in the Discussion:

Lines (379-380): An additional effect of the planning assessment from the multicenter study was more similar Dmean,SCR in the SCS-RT plans (Table S10).

  1. Limited Assessment of Patient-Reported Outcomes

The study heavily relies on NTCP models to estimate xerostomia risk without incorporating robust patient-reported outcomes or real-world data on quality of life, which could provide a more holistic view of the benefits.

Response: We acknowledge that we have only investigated the expected benefit of SCS radiotherapy using xerostomia NTCP models. Considering the design of the study, no follow-up data was available. However, almost all NTCP models do predict patient-reported outcomes (lines 136-143).

Moreover, a previous study applying SCS radiotherapy demonstrated the role of dose to the SCR regions (Steenbakkers et al. 2022). In addition, if you would apply the NTCP models for the several xerostomia outcomes as used in the current study in the cohort of Steenbakkers et al., the patients treated with SCS radiotherapy have consistently lower NTCPs than patients treated with standard parotid gland sparing radiotherapy (Van Rijn-Dekker et al. 2023).

For quality of life, many side effects play a role, not only xerostomia. Xerostomia is still one of the most reported side effects after radiotherapy. So we believe that reducing the chance of developing xerostomia will eventually have a positive impact on quality of life.

We added this last section to the limitations in the Discussion: Lines 410-413

  1. Lack of Emphasis on Economic and Operational Feasibility

While the article claims that SCS-RT requires only small adjustments, it does not analyze the economic or operational feasibility of implementation, such as training requirements for staff or costs associated with updated protocols.

Response: Indeed, no economic or operational feasibility study was performed as part of the current Manuscript. However, recently our center implemented SCS radiotherapy as standard of practice. We have added some of our experiences to the Discussion:

Lines (387-391): Lastly, the implementation of a new planning strategy requires time (e.g., adjusting protocols, training staff and evaluating the changes made). From our own experience and the multicenter study, the time investment to adopt SCS-RT in your own standard of practice is limited since it is only the addition of one OAR (i.e., the SCR regions).

Comments on the Quality of English Language

The English could be improved to more clearly express the research.

Response: We have thoroughly evaluated the quality of English in the Manuscript. If the Reviewers or the Editor still believe the English should be improved, we could send it to an English editor for review.

Reviewer 2 Report

Comments and Suggestions for Authors

hello

thank you for an interesting paper

the title is sound

title matches abstract

abstract is well structured

used key words a sufficient

nothing to change

authors do present a paper on xerostomia in hnscc

used summary is nicely presented - nothing to correct

1.

introduction is sound, nothing to change

a clear study object is written

citations re good

2.

material and methods

I'm missing study inclusion and exclusion criteria

I'm missing some information on what types of oral scc cases were used

were there only N+ cancer patients requiring RTH-therapy?

no exclusion criteria are set - does secondary oral SCC patients were included in the study?

did only patients with primary cancer got involved?

perhaps add a chart from why only 30 patients were included

did the selected patients had any past surgeries or oncology treatment?

does the TNM staging changed the doses of rth and the methodology?

please add to materials and methods

3.

table 1 is OK, same figure 1

results are well presented

other tables and figures are well presented

presented results are very nicely written

I would compare results with RTH protocol and the type of cancer, its TNM and necessity for rth - did all patients had the same RTH-protocol regardless theirs TNM?>

4.

discussion Is OK

used citations are good

references are sound

nothing to change

presented limitations are well written

5.

final conclusions are ok - perhaps would need some adjustments when material/method section will be adjusted

references and tables/figures are suitable

thank you

please improve

Author Response

Comments and Suggestions for Authors

hello

thank you for an interesting paper

the title is sound

title matches abstract

abstract is well structured

used key words a sufficient

nothing to change

authors do present a paper on xerostomia in hnscc

used summary is nicely presented - nothing to correct

1.

introduction is sound, nothing to change

a clear study object is written

citations re good

2.

material and methods

I'm missing study inclusion and exclusion criteria

I'm missing some information on what types of oral scc cases were used

were there only N+ cancer patients requiring RTH-therapy?

no exclusion criteria are set - does secondary oral SCC patients were included in the study?

did only patients with primary cancer got involved?

perhaps add a chart from why only 30 patients were included

did the selected patients had any past surgeries or oncology treatment?

does the TNM staging changed the doses of rth and the methodology?

please add to materials and methods

Response: We agree with the Reviewer that the patient selection needs to be clarified more. The patients for the study population were selected to represent the general HNC population. Therefore, we included patients with squamous cell carcinoma in different locations (i.e., oropharynx, nasopharynx, hypopharynx, larynx and oral cavity), in which we aimed to include at least 5 patients per tumor location category. In addition, patients with different TN classifications (i.e., T1-4 and N0-3) were selected. However, all patients were scheduled for definitive radiotherapy (70 Gy to tumor targets and 54.25 Gy to elective targets) with or without concurrent systemic treatment. Exclusion criteria were postoperative radiotherapy, previous head and neck irradiation, other tumor locations than the pharynx, larynx and oral cavity, and no bilateral neck irradiation.

We have elaborated on the selection of patients in the Materials and Methods:

Lines 104-109: In total, 30 HNC patients with squamous cell carcinoma tumors originating from the nasopharynx, hypopharynx, oropharynx, larynx and oral cavity with varying TN-staging were selected to represent the general HNC population. All patients were scheduled for definitive radiotherapy with or without concurrent systemic treatment. Exclusion criteria were postoperative radiotherapy, previous head and neck irradiation, other tumor locations than those mentioned above, and no bilateral neck irradiation.

3.

table 1 is OK, same figure 1

results are well presented

other tables and figures are well presented

presented results are very nicely written

I would compare results with RTH protocol and the type of cancer, its TNM and necessity for rth - did all patients had the same RTH-protocol regardless theirs TNM?>

Response: Indeed, all patients received the same radiotherapy protocol (70 Gy in fractions of 2 Gy to tumor targets and 54.25 Gy in fractions of 1.55 Gy to elective targets). Consequently, no comparison regarding the radiotherapy schedule was made. We have clarified this in the Materials and Methods section:

Lines 161-164: All patients received the same RT fractionation schedule: 70 Gy in fractions of 2 Gy to the tumor volume and 54.25 Gy in fractions of 1.55 Gy to the elective volume using a simultaneous integrated boost technique.

4.

discussion Is OK

used citations are good

references are sound

nothing to change

presented limitations are well written

5.

final conclusions are ok - perhaps would need some adjustments when material/method section will be adjusted

references and tables/figures are suitable

thank you

please improve

Round 2

Reviewer 1 Report

Comments and Suggestions for Authors

accept

Comments on the Quality of English Language

improve